# Multi-trait selection for drought-tolerant soybean accessions under contrasting water regimes

Tenena Silue[1,2]*, Bunmi Olasanmi[3], Adeyinka Saburi Adewumi[2], Paterne Angelot Agre[2], Idris Ishola Adejumobi[2], Paul Olabode Kehinde[2], Godfree Chigeza[4], Abush Tesfaye Abebe[2]*

1 Pan African University Life and Earth Science Institute, University of Ibadan, Ibadan, Nigeria, 2 International Institute of Tropical Agriculture (IITA), Ibadan, Nigeria, 3 Department of Crop and Horticultural Sciences, University of Ibadan, Ibadan, Nigeria, 4 International Institute of Tropical Agriculture, Southern Africa Research and Administration Hub (SARAH) Campus, Lusaka, Zambia

* at.abebe@cgiar.org (ATA), siluetenenan@yahoo.fr (TS)

## Abstract

Soybean, a crucial source of protein and oil, faces significant yield reductions due to water stress, particularly at the flowering and pod-filling stages. This study evaluated 150 accessions from the Soybean Breeding Program of the International Institute of Tropical Agriculture (IITA) in Ibadan, Nigeria, under both well-watered and water-stressed conditions across two growing seasons using a multi-trait selection. Intermittent water stress was imposed at critical reproductive stages (35 and 63 DAS). Significant genetic variation (p < 0.001) was observed for all traits under bothwater regimes.Broad-sense heritability estimates ranged from 21.56% to 86.56%, with the highest values observed for lodging score and days to 50% flowering.. Grain yield under stress correlated positively with yield in non-stress conditions (r = 0.38), while hundred-seed weight showed consistent positive associations with yield across water regimes. The multi-trait genotype-ideotype distance index (MGIDI) identified 30 superior accessions under each water regime, with eight consistently outstanding across both. Drought tolerance indices further supported the identification of high-yielding, resilient genotypes, with SY150 emerging as the most stable across all three selection approaches. These findings demonstrate that integrating MGIDI with drought tolerance indices provides a powerful approach for simultaneous selection of yield potential and stress resilience. The identified accessions represent promising candidates advancing water-deficit-tolerant soybean breeding in sub-Saharan Africa.

## Introduction

Soybean (*Glycine max* (L.) Merril; 2n = 20), often dubbed as the "Golden Bean" or the "Miracle Crop" of the 20th century, is the most widely cultivated seed legume across the world [1]. Renowned for its substantial protein and vegetable oil content, soybean is a crucial resource for human consumption and livestock feed [2]. Economically, it

**Data availability statement:** We appreciate your feedback regarding the Data Availability Statement. In accordance with the journal's data sharing policy, we have revised the statement to provide open and institutional access to the dataset. The revised statement now reads as follows: All relevant data are included within the manuscript and its Supporting Information files. Additionally, the phenotypic raw data and best linear unbiased estimates (BLUEs) used for the multi-trait genotype–ideotype distance index (MGIDI) analysis are available on the Figshare repository at https://doi.org/10.6084/m9.figshare.30529667.

**Funding:** The field experiment was funded Bill and Melinda Gates Foundation, grant number: Inv-046815-2023. The funds were received by Abush Tesfaye Abebe. The funders of this manuscript Bill and Melinda Gates Foundation had no role in the study design, data collection and analysis, decision to publish, or preparation of the manuscript.

**Competing interests:** The authors have declared that no competing interests exist.

is a major leguminous crop worldwide and holds a leading position among oilseed crops on the international market, surpassing others such as cottonseed, peanut (groundnut), sunflower seed, rapeseed, palm oil, and coconut [3]. Climate change is increasingly influencing agriculture, with drought being one of the most damaging abiotic stresses for crops, including soybean, particularly in semi-arid and arid regions [4,5]. Drought can have a variety of detrimental effects on plants, including the inhibition of photosynthesis [6], increased oxidative damage [7], and metabolic disruptions [8]. These effects result in reduced leaf area, plant height, pod and seed yield, seed weight and harvest index [9]. Previous studies have shown that drought can reduce soybean yields by approximately 40% [10] under moderate conditions, while in severe cases, yield losses may reach 100% [11].

Improving drought tolerance in soybeans is essential for minimizing yield losses and maintaining stable production during periods of drought. Understanding morphological, agronomic, and physiological responses to drought can help enhance drought resilience [12]. The impact of drought depends on its duration, intensity, and especially its timing within the growth cycle of the plant [5]. For soybeans, the flowering stage and the period right after are the most sensitive to water stress [9–12]. Therefore, assessing how different soybean varieties respond to drought during the critical growth stages is key to developing more tolerant cultivars.

Several drought-tolerant indices such as the stress susceptibility index (SSI), stress tolerance (TOL), mean productivity (MP), stress tolerance index (STI), yield stability index (YSI), geometric mean productivity (GMP), yield index (YI), and harmonic mean (HM) have been developed used to evaluate crop performance under both stress and well-watered conditions, each emphasizing different aspects of yield response. These indices have been widely applied in soybeans [13,14] and in other crops, including wheat [15–17], rice [18,19], maize [19–21], and bread wheat [22].

Though several selection indices have been developed for use in soybean improvement, all of them have focused solely on yield. However, it is a highly complex and polygenic trait, influenced by numerous underlying physiological, morphological, and developmental components [23]. Therefore, selecting solely on yield can be inefficient as favorable alleles for these underlying traits may be missed or unfavorably linked. In addition, yield performance is often highly variable across different environments (years, locations, stress levels, etc.). Genotypes with high yield in one environment might perform poorly in another. Focusing only on yield in a specific condition might lead to the selection of genotypes that are not broadly adapted or resilient. When dealing with conditions like water stress, yield is often a consequence of how well a plant manages the stress. Selecting for traits directly related to stress tolerance alongside yield can lead to genotypes that are better adapted to the specific challenge and exhibit more stable performance under stress [24].

Multi-trait selection allows for the identification of genotypes that not only yield well under optimum conditions but also maintain reasonable yields under water stress. This enhances overall stability and reduces the risk of significant yield losses in unfavorable years. It also enables breeders to incorporate traits valued by consumers

and industries, enhancing the economic relevance of selected varieties [24,25]. The multi-trait genotype-ideotype distance index (MGIDI), introduced by Olivoto & Nardino [26], is a promising approach that addresses the shortcomings of traditional yield-based methods by integrating the information of diverse traits to guide more balanced and genetically informed selection decisions [27]. It works by defining an "ideotype", a hypothetical genotype with the most desirable value for each trait, and then computing the distance of each genotype from this ideotype. Genotypes with shorter distances are considered superior because they combine multiple desirable traits simultaneously. For breeders, this makes MGIDI not just a statistical tool, but a practical decision-support framework that highlights both the strengths and weaknesses of candidate lines, guiding targeted improvement [28]. Although MGIDI has been applied in crops such as yams [29,30], maize [20,31], wheat [32], its use in soybean improvement under water-limited environments in sub-Saharan Africa remains largely unexplored. Yet, given the growing importance of soybeans in the region, the absence of systematic studies applying MGIDI for drought tolerance in tropical germplasm represents a critical gap. Addressing this gap is essential for developing stress-resilient varieties adapted to African agro-ecologies. By integrating MGIDI with traditional drought tolerance indices, this study presents a novel framework for soybean improvement. While drought indices enable targeted evaluation of yield stability under stress, MGIDI integrates multiple traits simultaneously, offering a more balanced and genetically informed process. This combined approach improves the accuracy of identifying genotypes that are both high-yielding and resilient under water regimes.

In this context, we presumed that significant variation exists among tropical soybean genotypes in response to drought stress, and that integrating multi-trait selection through MGIDI with traditional drought tolerance indices can effectively identify genotypes with superior yield stability and drought resilience under contrasting water regimes.

The IITA germplasm represents a particularly valuable resource for this type of study. It encompasses a wide range of genetic diversity adapted to tropical environments in sub-Saharan Africa, where water limitation is a recurrent constraint [33–35]. Evaluating this germplasm using MGIDI alongside drought tolerance indices offers unique insights for breeding resilient soybean tailored to African agro-ecologies.

In this context, this study aimed to assess 150 soybean accessions under contrasting water regimes in Nigeria. Specifically, the objectives were to:

1. Evaluate genotype performance across contrasting water regimes;

2. Examine the relationships among grain yield, drought tolerance indices, and key agronomic traits;

3. Predict selection responses using a multi-trait approach, and

4. Identify superior and stable accessions suitable for both water-stressed and well-watered conditions.

## Materials and methods

### Experimental material and experimental site

A total of 150 soybean accessions obtained from the International Institute of Tropical Agriculture (IITA) soybean breeding program (S1 Table in S1 File) were evaluated under water-stressed (WS) conditions during the dry seasons of 2022–2023 and 2023–2024, as well as under well-watered conditions during the rainy seasons of 2023 and 2024. These trials were conducted at the IITA experimental station in Ibadan, Nigeria, situated at 07° 49´ N latitude, 3° 90´E longitude, and an elevation of 212 m a.s.l., located in the forest-Savannah transition zone. The region experiences a bimodal rainfall pattern with an annual total of 1305 mm. The soil of the IITA station is derived from metamorphic rocks of the Pre-Cambrian Complex, which include banded gneiss, quartzites, quartz schists, and granitic gneisses. The surface layer consists of sandy loam, underlain by a gravelly, clay-rich subsoil, and is classified as Alfisols [30; 31]. Field capacity (FC) of the surface soil (0–15 cm) at the IITA experimental station is approximately 18% soil moisture content measured at 20 kPa, whereas the permanent wilting point (PWP), determined at a matric potential of −1500 kPa,

is approximately 9% [36]. Across the two experimental years, distinct differences were observed in rainfall and temperature patterns between water regimes. Under water-stressed conditions, total rainfall ranged from 0 to 74.7 mm (average 25.6 mm) in the first year (December 2022-March 2023) and from 0 to 109.6 mm (average 50.8 mm) in the second year (December 2023-March 2024) (S1 Fig in S1 File). The corresponding minimum and maximum temperatures averaged 22.1 °C and 33.2 °C in the first year, and 23.7 °C and 34.1 °C in the second year, respectively (S2 Fig in S1 File). In contrast, under well-watered conditions, rainfall ranged from 99.1 to 310.8 mm (average 237.4 mm) in the first year (July-October 2024) and from 132.9 to 327.2 mm (average 228.6 mm) in the second year (July-October 2024) (S1 Fig in S1 File). The minimum temperature averaged 22.95 °C with a range between 22.7 °C and 23.2 °C in the first year (July-October 2023), and varied from 22.6 °C to 23.1 °C (average 22.82 °C) in the second year (July-October 2024), while maximum temperatures ranged from 28.7 °C and 30 °C (average 29.1°C) in the first year, and from 27.5 °C to 33.4°C in the second year (S2 Fig in S1 File).

## Experimental design and field management

The experiments were laid out using a 10 x 15 alpha lattice design with three replications. Each replication consisted of 10 blocks, and each block contained 15 accessions, giving a total of 150 experimental plots per replication. Each experimental plot consisted of two rows, each 2.0 meters in length and spaced 50 centimeters apart, with a sowing density of 20 plants per meter. The plot was considered the experimental unit. Accessions were randomly assigned to plots within each block, and the lattice structure was used to reduce field heterogeneity. In addition, block and replication effects were included in the statistical model to correct for spatial variation. Two contrasting water regimes were applied: well-watered (WW) during the rainy season and water-stressed (WS) conditions during the dry season. Under WS conditions, plants received 17 mm of sprinkler irrigation water for 3 hours, three times a week, up to 35 days after sowing (DAS). At 35 DAS, a 14-day drought period was imposed, followed by 14 days of irrigation. A second water stress was initiated at 63 DAS, lasting 7 days, after which irrigation resumed until the plants reached maturity. These two stress periods coincided with the flowering and pod-filling stages, respectively.

The irrigation was achieved using a portable overhead sprinkler system available at IITA, Ibadan, Nigeria. The system comprises impact-type sprinklers (Rain Bird 35A) mounted on galvanized riser pipes and connected through PVC laterals, spaced at 12 × 12 m and operated at 200–250 kPa, ensuring uniform water distribution across the field.

The timing and duration of the stress treatments were guided by field realities in Nigeria, where soybean production is frequently constrained by the recurrent "August break", an intermittent drought characterized by a cessation of rainfall during the rainy season, which can take from one to three weeks. This water deficit period generally coincides with the flowering or pod filling stages of soybean, which are recognized as the most drought-sensitive growth and yield production [37]. Therefore, to mimic these field conditions, a two-week water deficit period was imposed at flowering and a one-week water stress period at pod filling, giving a total of 21 days of water deficit periods.

In addition to the different irrigation treatments, all field management practices were consistently applied across both well-watered and water-stressed experiments. A combination of fertilizers, including NPK 15:15:15 and TSP, was applied at the rates of 100 kg/ha and 150 kg/ha, respectively. Soybean seeds, pre-inoculated with Nodumax, a commercial soybean inoculant produced by IITA, containing *Bradyrhizobium japonicum* strain, were sown manually by drilling the seeds in the rows. This inoculation aimed to enhance nodulation capacity and improve nitrogen fixation in the soybean accessions. Thinning was done at 14 DAS to maintain a 5 cm spacing between plants, equivalent to 400,000 plants/ha. Pre-emergence weed herbicide mixture containing glyphosate (360 g/L), metribuzin (135 g/L), and S-metolachlor (405 g/L) was sprayed at a rate of 5 L/ha to control weeds. Top-dressing of urea (46% N) fertilizer was made at the rate of 50 kg/ha at 21 DAS, and manual hand-weeding was carried out as needed for additional weed control.

## Data collection

In each experiment, data on days to 50% flowering (D50F), days to 95% maturity (D95M), plant height (PH), fresh biomass (FB), number of pods per plant (NPP), number of seeds per pod (NSPP), lodging score (LS), hundred seed weight (HSW), and grain yield (GY) were collected as detailed in S2 Table in S1 File. Although some traits (e.g., PH, NPP, NSPP) were measured on a subset of individual plants within each plot, their values were averaged to generate a single plot mean.

To assess drought tolerance among the accessions, eight indices (equations 1–8) were computed for each genotype using grain yield recorded under water-stressed (Ys) and well-watered (Yp) conditions across the two years of evaluation. The indices were calculated based on the following est:

These indices were based on the grain yield recorded under both drought and optimal conditions in each planting year, using the following formula [38–43]:

$$\text{Stress Tolerance Index (STI)} = \frac{(Y_s)\,(Y_P)}{\bar{y}_p * \bar{y}_p} \tag{1}$$

$$\text{Geometric Mean Productivity (GMP)} = \sqrt{(Ys)(Yp)} \tag{2}$$

$$\text{Mean Productivity (MP)} = \frac{Y_S + Y_P}{2} \tag{3}$$

$$\text{Harmonic mean (HM)} = \frac{2(Yp * Ys)}{Yp + Ys} \tag{4}$$

$$\text{Tolerance Index (TO)} = Yp - Ys \tag{5}$$

$$\text{Stress susceptibility index (SSI)} = \frac{\left[1 - \left(\frac{Y_S}{Y_P}\right)\right]}{SI}; \ \text{Stress Intensity } (SI) = [1 - (\bar{y}s)/(\bar{y}p)] \tag{6}$$

$$\text{Yield Stability Index (YSI)} = \frac{Y_s}{Y_P} \tag{7}$$

$$\text{Yield index (YI)} = \frac{Y_S}{\bar{y}_S} \tag{8}$$

These drought tolerance indices capture complementary aspects of genotype response to water stress. MP reflects the overall yield potential across environments rather than specific drought tolerance [39]. GMP, HM, and STI emphasize stable and high-yield performance under both stress and non-stress conditions and are therefore strong indicators of drought tolerance [38,40]. YI and YSI assess relative performance and yield stability under stress, reflecting adaptation to water-limited environments [42,43]. In contrast, TOL and SSI quantify yield reduction due to stress, with higher values indicating susceptibility rather than tolerance [39,41].

## Data analysis

Analysis of variance (ANOVA) was computed for each water condition using the mixed linear model (MLM), implemented via the lmerTest package [44] in R programming language version 4.4.2 [45]. In this model, the research conditions

(water-stressed or well-watered) and the year were considered as environments for the combined analysis. This resulted in two environments for each of the water regimes: WW_23 (well-watered 2023), WW_24 (well-watered 2024), WS_23 (water stress 2023), and WS_24 (water stress 2024). The environment, block, and replications were treated as random effects, while genotype was considered as a fixed effect, as shown in equation 9:

$$Y = \mu + Rep + Rep\,(Blk) + G + E + G\,x\,E + e \tag{9}$$

Where: Y = phenotype; μ = mean of the trait; G = genotype; E = environment; Rep = replication; Rep (Blk) = replication nested in the block; G x E = genotype by environment interaction; e = residual.

Best linear unbiased estimates (BLUEs) were computed for the grain yield and other agronomic traits across the tested environment for each water regime using the MLM. In addition, the coefficient of variation (CV) and broad-sense heritability (H²) were estimated using the procedure as described by Sansa et al. [46]. This model considers accessions and other variables, including environment, replication and block as random effects.

$$H^2 = \frac{\delta_g^2}{\delta_g^2 + \frac{\delta_{ge}^2}{nEnv} + \frac{\delta_\varepsilon^2}{(nEnvs\ x\ nRep)}} \tag{10}$$

Where $\delta_g^2$, $\delta_{ge}^2$ and $\delta_\varepsilon^2$ represent the genotypic, genotype x environment interaction, and the error variance components, respectively, nRep is the number of replications, and nEnvs is the number of environments.

As reported by Osman & Khidir [47] the heritability estimates were classified into three categories: low (less than 30%), moderate (between 30% and 60%), and high (greater than 60%).

The yield estimates for each genotype, derived from the BLUEs, were used to calculate the eight drought tolerance indices (DTI) for each water regime. To identify the genotype with the highest drought tolerance, a ranking analysis was performed based on each index value, where accessions were ranked according to their performance for each index. The genotype with the highest performance received a rank of 1, while the lowest-ranked genotype was assigned a rank of 150 [14]. To identify drought-tolerant accessions, a rank sum (RS) [48] was calculated using the following formula:

$$Rank\ sum\ (RS) = Rank\ mean\ (R) + Standard\ deviation\ of\ rank\ (SDR) \tag{11}$$

The standard deviation of ranks (SDR) was calculated as:

$$SDR = \sqrt{S_i^2} = \frac{\sum_{i=1}^{m} (Rij - Ri)^2}{n-1} \tag{12}$$

Accessions with the lowest RS were classified as drought-tolerant, while those with the highest RS were the most susceptible to drought. To further refine the selection of the accessions, a biplot analysis was performed using the GGEBiplotGUI package in R [49]. This analysis highlighted the interaction between the accessions and the different drought tolerance indices [20]. In addition, Pearson's correlation analyses were performed to examine the relationships between the drought tolerance indices and the genotypic mean yields across the two years of evaluation, as well as the correlation between the agronomic traits and the grain yield under each water condition, using the metan package v.1.18 [50].

The MGIDI distance index, proposed by [26], was applied to identify accessions that effectively integrate most of the evaluated traits within each environment in an ideal way. This approach involved determining the optimal genotype and rescaling the variables to a 0–100 scale using the following equation [26]:

$$rX_{ij} = \frac{\eta_{nj} - \phi_{nj}}{\eta_{oj} - \phi_{oj}} \chi\ (\theta_{ij} - \eta_{oj}) + \eta_{nj} \tag{13}$$

Where $\eta_{nj}$ and $\varphi_{nj}$ represent the new maximum and minimum values for trait j, respectively, after rescaling, while $\eta_{oj}$ and $\varphi_{oj}$ are the original maximum and minimum values for trait j. $\theta_{ij}$ denotes the original value of the j-th trait for the i-th genotype.

For traits such as D50F, D95M, and LS, where lower values are preferred, $\eta_{nj}$ is set to 0 and $\varphi_{nj}$ to 100. For other traits where higher values are desired, $\eta_{nj}$ is set to 100 and $\varphi_{nj}$ to 0. After rescaling, the optimal genotype was represented by a value of 100 for all the traits.

Next, exploratory factor analysis was conducted using $rX_{ij}$ to categorize the related traits and reduce the data's dimensionality. This process generated factor loadings for each genotype, as described in the following equation [26].

$$X = \mu + Lf + \varepsilon \tag{14}$$

X is a 1p x 1 vector representing the rescaled observations, while μ is a 1p x 1 vector indicating the standardized means. The vector f, with dimensions 1px1, represents the common factors, and ε is a 1p x 1 vector of residuals. In this formula, p and f, respectively, highlight the number of retained traits and the number of common factors. The eigenvalues and eigenvectors were calculated from the correlation matrix of $rX_{ij}$, with only those having eigenvalues greater than 1 being retained. The scores were then computed using the following equation:

$$F = Z(A^T R^{-1})^T \tag{15}$$

Where F is a g x f matrix that holds the factorial scores, while Z is a g x p matrix containing the standardized means. A is a p x f matrix representing the canonical loadings, and R is a p x p correlation matrix among the traits. In this context, g,f and p represent the number of accessions, the number of retained factors, and the number of analyzed traits, respectively.

Following this, the Euclidean distance between genotype scores and the ideal accessions was calculated following the MGIDI index formula:

$$MGIDI = \left[\sum_{j=1}^{f}(Y_{ij} - Y_j)^2\right]^{1/2} \tag{16}$$

$Y_{ij}$ represents the score of the i-th genotype on the j-th factor (i = 1, 2,...., g; j = 1, 2,...., f), where g and f refer to the number of accessions and factors, respectively. $Y_j$ denotes the scores of the j-th ideal genotype.

The genotype with the smallest MGIDI is closest to the ideal genotype, exhibiting the most favorable values for all assessed traits. The selection differential for all traits was calculated with a selection intensity of 20%. As a result, accessions with lower MGIDI values, meaning those that are nearer to the ideal genotype, were selected. Subsequently, the proportion of the MGIDI explained by the correlated factor was used to highlight the strengths and weaknesses of the accessions, as shown in the equation below:

$$\omega_{ij} = \frac{\sqrt{D_{IJ}^2}}{\sum_{i=1}^{f}\sqrt{D_{ij}^2}} \tag{17}$$

Where $\omega_{ij}$ represents the proportion of the MGIDI of the i-th genotype that is explained by the correlated j-th factor, while D2ij denotes the distance between the i-th genotype and the ideotype for the j-th factor.

## Results

### Analysis of variances and broad-sense heritability

The mean square values from the combined analysis of variance (ANOVA) and broad-sense heritability estimates ($H^2$) for grain yield and agronomic traits under the water-stressed and well-watered conditions are presented in Tables 1 and 2,

respectively. Highly significant differences (p < 0.001) in genotype mean square were observed for traits across the two water regimes. Highly significant (p < 0.001) genotype by environment interaction (GEI) was found for all the traits under water-stressed conditions, except for fresh biomass (FB) and lodging score (LS) (Table 1). Under well-watered conditions, only days to 50% flowering (D50F), days to 95% maturity (D95M), and grain yield (GY) showed highly significant (p < 0.01) genotype x environment interaction (Table 2). The average grain yield under water-stressed conditions was 859.5 kg/ha, with values ranging from 252.5 kg/ha to 1680.7 kg/ha, while under well-watered conditions, it ranged from 850.1 kg/ha to 3651.2 kg/ha, with an average grain yield of 2324.3 kg/ha. The coefficient of variation (CV) for the well-watered conditions ranged from 3.1% for D95M to 44.9% for lodging score (LS), while under water-stressed conditions, it ranged from 2.6% for D95M to 36.5% for LS. The $H^2$ estimates under water-stressed conditions ranged from a low heritability of 28.6% for hundred-seed weight (HSW) to a high heritability of 86.6% for LS, while under well-watered conditions ranged from 21.6% for GY to 81.6% for D50F (Tables 1 and 2).

**Table 1. Mean squares, coefficient of variations and heritability of grain yield and measured agronomic traits under water-stressed conditions.**

| Source of variation | Df | D50F | D95M | FB | PH | NPP | NSPP | LS | HSW | GY |
|---|---|---|---|---|---|---|---|---|---|---|
| Env | 1 | 1353.8*** | 1568.63*** | 119.97 ns | 4.07 ns | 11.3 ns | 2.58*** | 0.00025 ns | 25.96** | 577662*** |
| Gen | 149 | 114.22*** | 104.30*** | 588.64*** | 601.11*** | 4372.9*** | 0.075*** | 4.91*** | 7.06*** | 4345698*** |
| Env x Gen | 149 | 25.81*** | 64.04*** | 32.60 ns | 169.05*** | 1860.6*** | 0.052*** | 0.33 ns | 5.05*** | 331846*** |
| Residual | | 1.64 | 2.82 | 12.5 | 8.05 | 33.78 | 0.34 | 0.81 | 0.60 | 185.55 |
| Min | | 35 | 101 | 15.05 | 42.66 | 59.00 | 2.03 | 1.0 | 7.24 | 252.54 |
| Mean | | 44 | 109 | 34.38 | 62.11 | 114.7 | 2.31 | 2.2 | 12.12 | 859.49 |
| Max | | 55 | 120 | 61.15 | 90.16 | 221.67 | 2.67 | 4.7 | 15.86 | 1680.75 |
| CV (%) | | 3.74 | 2.58 | 36.24 | 12.98 | 29.46 | 7.78 | 36.53 | 5.01 | 21.52 |
| $H^2$ (%) | | 77.45 | 38.45 | 74.20 | 71.99 | 57.54 | 30.24 | 86.56 | 28.56 | 42.97 |

*, **, *** Significant at p < 0.05, p < 0.01, and p < 0.001, respectively; ns: non-significant, Df: degree of freedom, D50F: days to 50% flowering, D95M: days to 95% maturity, FB: fresh biomass, PH: plant height, NPP: number of pods per plant, NSPP: number of seeds per pods, LS: lodging score, HSW: hundred seed weight, GY: grain yield, CV: coefficient of variation, $H^2$: broad-sense heritability

**Table 2. Mean squares, coefficient of variations and heritability of grain yield and measured agronomic traits under well-watered conditions.**

| Source of variation | Df | D50F | D95M | FB | PH | NPP | NSPP | LS | HSW | GY |
|---|---|---|---|---|---|---|---|---|---|---|
| Env | 1 | 89.88* | 835.47*** | 31.89 ns | 8.36 ns | 81.21 ns | 0.11 ns | 0.23 ns | 10.71 ns | 204477600.5*** |
| Gen | 149 | 70.85*** | 105.63*** | 7087.96 *** | 643.40*** | 3605.50*** | 0.21 *** | 5.02*** | 9.52*** | 1548986.61*** |
| Env x Gen | 149 | 13.12** | 29.58*** | 137.80 ns | 221.60** | 1674.57 ns | 0.07 ns | 0.27 ns | 2.33 ns | 1217437.63*** |
| Residual | | 2.98 | 3.28 | 40.56 | 12.61 | 38.9 | 0.25 | 1.04 | 1.38 | 715.53 |
| Min | | 35 | 96 | 52.27 | 46.79 | 69 | 2.16 | 1.0 | 11.03 | 850.06 |
| Mean | | 43 | 105 | 121.52 | 71.04 | 129 | 2.59 | 2.33 | 14.07 | 2324.26 |
| Max | | 55 | 116 | 235.60 | 113.04 | 195 | 3.54 | 4.45 | 17.17 | 3651.21 |
| CV (%) | | 6.92 | 3.12 | 33.37 | 17.75 | 30.24 | 9.76 | 44.89 | 9.82 | 30.67 |
| $H^2$ (%) | | 81.67 | 72.51 | 77.71 | 66.46 | 54.70 | 67.98 | 78.93 | 76.30 | 21.56 |

*, **, *** Significant at p < 0.05, p < 0.01, and p < 0.001, respectively; ns: non-significant, Df: degree of freedom, D50F: days to 50% flowering, D95M: days to 95% maturity, FB: fresh biomass, PH: plant height, NPP: number of pods per plant, NSPP: number of seeds per pods, LS: lodging score, HSW: hundred seed weight, GY: grain yield, CV: coefficient of variation, $H^2$: broad-sense heritability

## Drought tolerance indices and genotype ranking

Drought tolerance assessment using multiple indices showed that relying on a single criterion was insufficient due to inconsistent genotype rankings. To enhance selection accuracy, mean rank, standard deviation of rank (SDR), and rank sum (RS) were calculated across all indices, with 30 accessions selected under a 20% selection intensity (SI) threshold (Table 3; S3 Table in S1 File). As presented in Table 3, genotype SY010 emerged as the most drought-tolerant, with the lowest RS (23.41) and SDR (13.61), and ranking highest in Ys and YI, though it showed lower performance under non-stress conditions (Yp) and TOL, ranking 44th and 22nd, respectively. Accessions SY032, SY095 also exhibited low RS values (28.29 and 31.25, respectively), were consistently high ranked across multiple indices. They showed a balanced performance under both water-stressed and well-watered conditions. SY101 and SY043, respectively, ranked 4th and 5th overall, also demonstrated balanced performance and low variability across the different indices. In contrast, accessions like SY052, SY094, SY104, and SY100 showed moderate performance for some indices, and excelled in some indices. Some top-30 accessions, such as SY147, SY053, SY099, and SY112, were identified as drought-sensitive due to high RS. They have low rankings in Ys, YSI, STI, and elevated TOL and SSI values. The GGE biplot analysis (Fig 1A) accounted for 95.9% of the total variation, with axis I and axis II explaining 59.36% and 36.55%, respectively. Indices such as STI, GMP, MP, and Ys were clustered closely together (angle < 90°), suggesting a strong positive correlation among them. In contrast, TOL and SSI showed a positive correlation with each other but were positioned at wider angles (> 90°) from indices like YSI and Ys, indicating a negative relationship with those indices. Accessions positioned near specific indices have high performance in those indices. Hence, SY010 aligned closely with Ys and YI, reaffirming its top ranking as the most drought-tolerant genotype. SY032 and SY095 clustered near STI, GMP, and HM, confirming their consistent and balanced performance under both water regimes. Accessions SY005 and SY052 aligned with YSI, indicating superior yield stability under stress conditions. In contrast, accessions such as SY099, SY053, and SY147 were located near TOL and SSI indices, highlighting their sensitivity and poor performance under drought conditions. Meanwhile, SY094 and SY117 were closely associated with Yp and MP. Additionally, the "Which Won Where/What" polygon view (Fig 1B) further classified the selected accessions into eight sectors, four of which contained relevant drought indices.

## Phenotypic correlation between drought tolerance indices and other agronomic traits under water-stressed and well-watered conditions

Fig 2 illustrates the relationships among drought tolerance indices, mean grain yield under both well-watered (Yp) and water-stressed (Ys) conditions, and the correlations between grain yield (GY) and key agronomic traits under the two water regimes. A significant positive correlation was found between Yp and Ys ($r = 0.38$, $p < 0.001$), indicating consistency in genotype performance across environments. Yp was significantly ($p < 0.05$) and positively correlated with all drought tolerance indices, except for the yield stability index (YSI). In contrast, Ys was significantly ($p < 0.05$) negatively correlated with both YSI and the tolerance index (TOL), while showing positive and significant correlations with the remaining indices (Fig 2A). Under water-stressed conditions, GY exhibited a highly significant positive correlation with hundred-seed weight (HSW; $r = 0.25$, $p < 0.01$) and number of seeds per pod (NPP; $r = 0.21$). Conversely, a highly significant negative correlation was observed between GY and days to 95% maturity (D95M; $r = -0.28$, $p < 0.001$). Furthermore, D95M showed a strong positive correlation with days to 50% flowering (D50F; $r = 0.61$, $p < 0.001$), while both D95M and D50F were significantly negatively correlated with HSW ($r = -0.38$ and $r = -0.30$, respectively; $p < 0.001$) (Fig 2B). Under well-watered conditions, GY showed significant positive correlations with HSW ($r = 0.19$; $p < 0.05$) and NPP ($r = 0.26$, $p < 0.01$) (Fig 2C).

## Multi-trait genotype-ideotype distance index selection and predicted selection gains

A total of 30 accessions were selected based on the MGIDI method, evaluated separately under each water regime, with a selection intensity of 20% (Fig 3). During water-stressed conditions, the MGIDI analysis grouped the evaluated traits into

**Table 3. Drought tolerance indices rank, rank mean (R), standard deviation of ranks (SDR) and rank sum (RS) of the selected soybean accessions.**

| Gen | Ys | Yp | STI | GMP | MP | TOL | HM | YSI | YI | SSI | R | SDR | RS |
|---|---|---|---|---|---|---|---|---|---|---|---|---|---|
| SY010 | 1 | 44 | 2 | 2 | 7 | 22 | 2 | 8 | 1 | 9 | 9.8 | 13.61209 | 23.41209 |
| SY032 | 2 | 39 | 6 | 3 | 8 | 29 | 3 | 23 | 4 | 28 | 14.5 | 13.78606 | 28.28606 |
| SY095 | 5 | 54 | 10 | 9 | 20 | 26 | 5 | 14 | 5 | 15 | 16.3 | 14.95215 | 31.25215 |
| SY101 | 9 | 48 | 13 | 11 | 19 | 42 | 9 | 17 | 10 | 19 | 19.7 | 13.94473 | 33.64473 |
| SY043 | 11 | 53 | 7 | 13 | 24 | 38 | 10 | 21 | 8 | 22 | 20.7 | 14.80278 | 35.50278 |
| SY098 | 12 | 55 | 22 | 17 | 27 | 41 | 11 | 16 | 11 | 16 | 22.8 | 14.55869 | 37.35869 |
| SY005 | 3 | 79 | 8 | 10 | 26 | 8 | 4 | 6 | 2 | 7 | 15.3 | 23.37639 | 38.67639 |
| SY150 | 10 | 22 | 3 | 6 | 9 | 71 | 7 | 26 | 6 | 25 | 18.5 | 20.31557 | 38.81557 |
| SY047 | 8 | 35 | 12 | 7 | 13 | 56 | 6 | 37 | 12 | 39 | 22.5 | 17.62101 | 40.12101 |
| SY052 | 6 | 84 | 29 | 23 | 41 | 11 | 12 | 5 | 9 | 3 | 22.3 | 24.833 | 47.133 |
| SY094 | 4 | 8 | 1 | 1 | 2 | 94 | 1 | 34 | 3 | 37 | 18.5 | 29.87846 | 48.37846 |
| SY013 | 16 | 75 | 33 | 30 | 46 | 30 | 21 | 27 | 16 | 29 | 32.3 | 17.38486 | 49.68486 |
| SY104 | 15 | 36 | 24 | 14 | 22 | 75 | 14 | 42 | 20 | 45 | 30.7 | 19.32787 | 50.02787 |
| SY100 | 17 | 16 | 14 | 12 | 11 | 98 | 15 | 25 | 14 | 23 | 24.5 | 26.20963 | 50.70963 |
| SY096 | 24 | 42 | 28 | 24 | 30 | 83 | 23 | 36 | 25 | 36 | 35.1 | 17.9966 | 53.0966 |
| SY011 | 27 | 72 | 34 | 39 | 52 | 46 | 33 | 33 | 27 | 33 | 39.6 | 13.84197 | 53.44197 |
| SY105 | 7 | 95 | 35 | 25 | 50 | 7 | 17 | 2 | 7 | 2 | 24.7 | 29.28424 | 53.98424 |
| SY119 | 29 | 49 | 32 | 31 | 38 | 77 | 28 | 46 | 30 | 47 | 40.7 | 15.07057 | 55.77057 |
| SY022 | 23 | 81 | 48 | 40 | 54 | 34 | 31 | 18 | 24 | 17 | 37 | 19.79338 | 56.79338 |
| SY075 | 20 | 29 | 9 | 19 | 21 | 93 | 19 | 52 | 17 | 55 | 33.4 | 25.81214 | 59.21214 |
| SY097 | 22 | 24 | 17 | 20 | 18 | 100 | 20 | 49 | 22 | 50 | 34.2 | 26.16529 | 60.36529 |
| SY099 | 19 | 23 | 16 | 15 | 17 | 96 | 18 | 55 | 21 | 59 | 33.9 | 27.19252 | 61.09252 |
| SY124 | 45 | 57 | 36 | 44 | 51 | 76 | 42 | 53 | 42 | 53 | 49.9 | 11.1997 | 61.0997 |
| SY045 | 33 | 41 | 40 | 29 | 35 | 87 | 27 | 54 | 41 | 58 | 44.5 | 17.92732 | 62.42732 |
| SY117 | 14 | 7 | 4 | 4 | 3 | 121 | 8 | 47 | 13 | 46 | 26.7 | 37.05267 | 63.75267 |
| SY142 | 34 | 45 | 31 | 32 | 36 | 86 | 30 | 61 | 34 | 64 | 45.3 | 18.88591 | 64.18591 |
| SY112 | 47 | 70 | 51 | 46 | 58 | 64 | 43 | 59 | 48 | 65 | 55.1 | 9.338689 | 64.43869 |
| SY092 | 36 | 47 | 15 | 34 | 40 | 84 | 32 | 63 | 31 | 66 | 44.8 | 20.49824 | 65.29824 |
| SY147 | 41 | 32 | 27 | 41 | 32 | 107 | 44 | 38 | 32 | 32 | 42.6 | 23.25797 | 65.85797 |
| SY053 | 35 | 86 | 62 | 49 | 66 | 35 | 41 | 35 | 45 | 38 | 49.2 | 17.04765 | 66.24765 |

Yp: yield under well-watered conditions; Ys: yield under water-stressed conditions; STI: stress tolerance index.; SSI: stress susceptibility index; YI: yield index; YSI: yield stability index; MP: mean productivity; TOL: tolerance index; GMP: geometric mean productivity; HM: harmonic mean, R = rank mean, SDR = standard deviation of ranks, RS = rank sum.

four factors with an average communality of 0.65 and uniqueness of 0.35. The first factor (FA1) was primarily associated with days to 50% flowering (D50F), days to 95% maturity (D95M), and hundred-seed weight (HSW). The second factor (FA2) was correlated with plant height (PH) and number of pods per plant (NPP), while the third factor (FA3) was linked to fresh biomass (FB) and lodging score (LS). The fourth factor (FA4) is associated with grain yield (GY) and number of seeds per pod (NSPP). All the measured traits showed favorable predicted selection gains, aligning with the overall breeding objectives. The MGIDI method predicted a cumulative selection gain of 68.1% for traits where increased expression was desirable, and a selection gain of −12.8% for traits where reduced expression was preferred (Table 4). During well-watered conditions, the MGIDI classified the measured traits into three factors. The first factor (FA1) was correlated with PH, NSPP, and LS, while the second factor (FA2) was associated with GY, D95M, and FB. The third factor (FA3) was

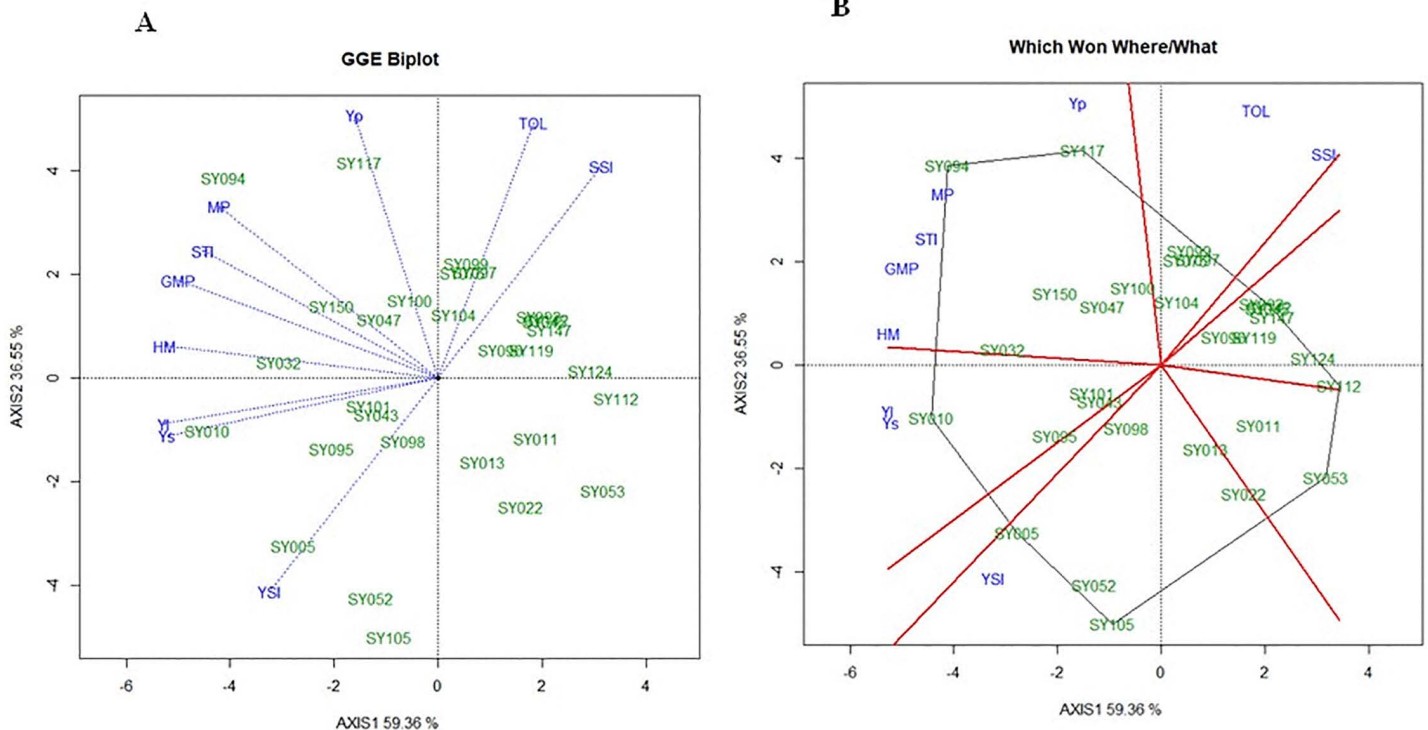

**Fig 1. Genotype by drought tolerance biplot (A) and polygon diagram biplot (B) of 150 soybean accessions across the two years of experimentation.**

correlated with D50F, NPP, and HSW. Among the nine traits evaluated, five showed desired genetic gain for the MGIDI index. However, four traits, D95M (0.34%), FB (−11.90%), NPP (−11.21%) and HSW (−1.83%), exhibited undesired selection gains. The average communality and uniqueness accounted for the respective 0.53 and 0.47 of the total variations in the studied soybean accessions (Table 5).

### Strengths and weaknesses of selected accessions using the MGIDI method

Figs 4A and 4B illustrate the strengths and weaknesses of selected accessions under water-stressed and well-watered conditions, respectively. The MGIDI contribution for each genotype was ranked from the most influential factor (FA) (located near the plot center) to the least influential factor (located farthest from the center). During water-stressed conditions, the traits were partitioned into four factors (Fig 4A, S4 Table in S1 File). Accessions SY021, SY026, SY078, SY091, SY093, SY099, SY116, and SY118 exhibited strengths associated with FA1, which is linked to D50F, D95M and GY. The accessions SY001, SY008, SY030, SY063, SY077, SY096, SY100, SY106, SY110 and SY113 showed strong associations to FA2, which was correlated with PH and NSPP. Accessions SY007, SY012, SY029, SY031, SY072, SY103, SY105, SY109 and SY146 demonstrated strong association with FA3, which was also associated with FB and LS. Finally, SY017, SY023 and SY150 showed strong associations to FA4, which is associated with HSW and NPP (Fig 4A, S4 Table in S1 File). Fig 4B highlights the strengths and weaknesses of the selected accessions under well-watered conditions, categorizing the traits into four factors. Only genotype SY149 exhibited strengths for FA1, which is linked to PH, NSPP and HSW. Accessions SY021, SY026, SY034, SY038, SY079, SY106, SY115, SY121, SY134, SY136, SY137, and SY143 showed strengths for FA2, which is associated with D50F and LS. The accessions SY017, SY018, SY019, SY020, SY022,

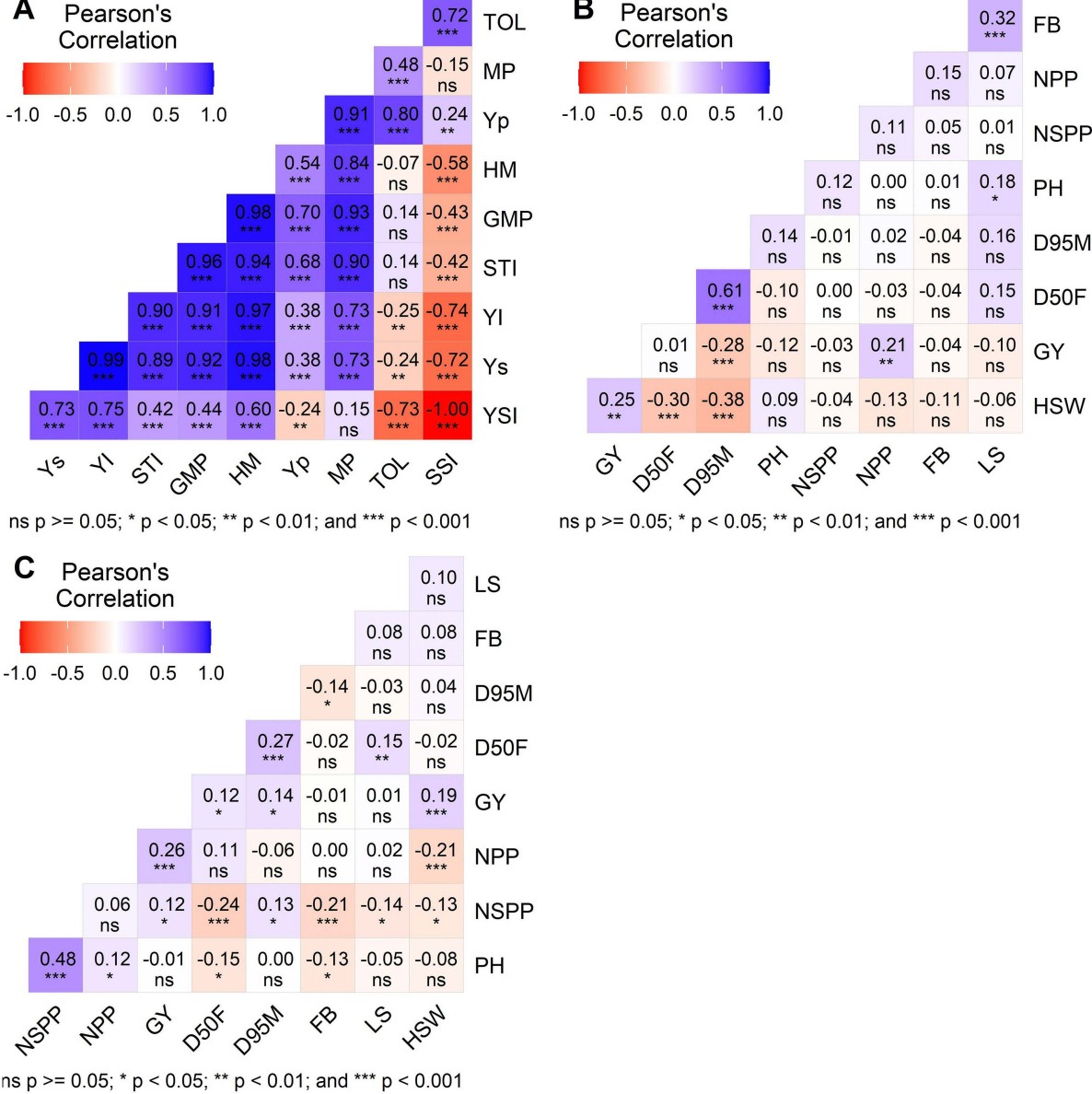

**Fig 2. Phenotypic correlation among grain yield, drought tolerance indices and agronomic traits under varying water availability.** (A) drought tolerance indices, (B) water-stressed conditions, (C) under well-watered conditions.

SY039, SY107, SY108, SY110, SY116, SY118, SY127, SY129, SY142, SY144, and SY150 demonstrated strengths for FA3, which is correlated with D95M and NPP. However, only SY071 was found to be associated with FA4, which is also associated with FB and GY (Fig 4B, S5 Table in S1 File).

## Venn diagram analysis

The Venn diagram comparing the three selection methods, including the RS ranking selection based on drought tolerance indices (DTI), and the MGIDI method under both water-stressed and well-watered conditions, revealed a substantial

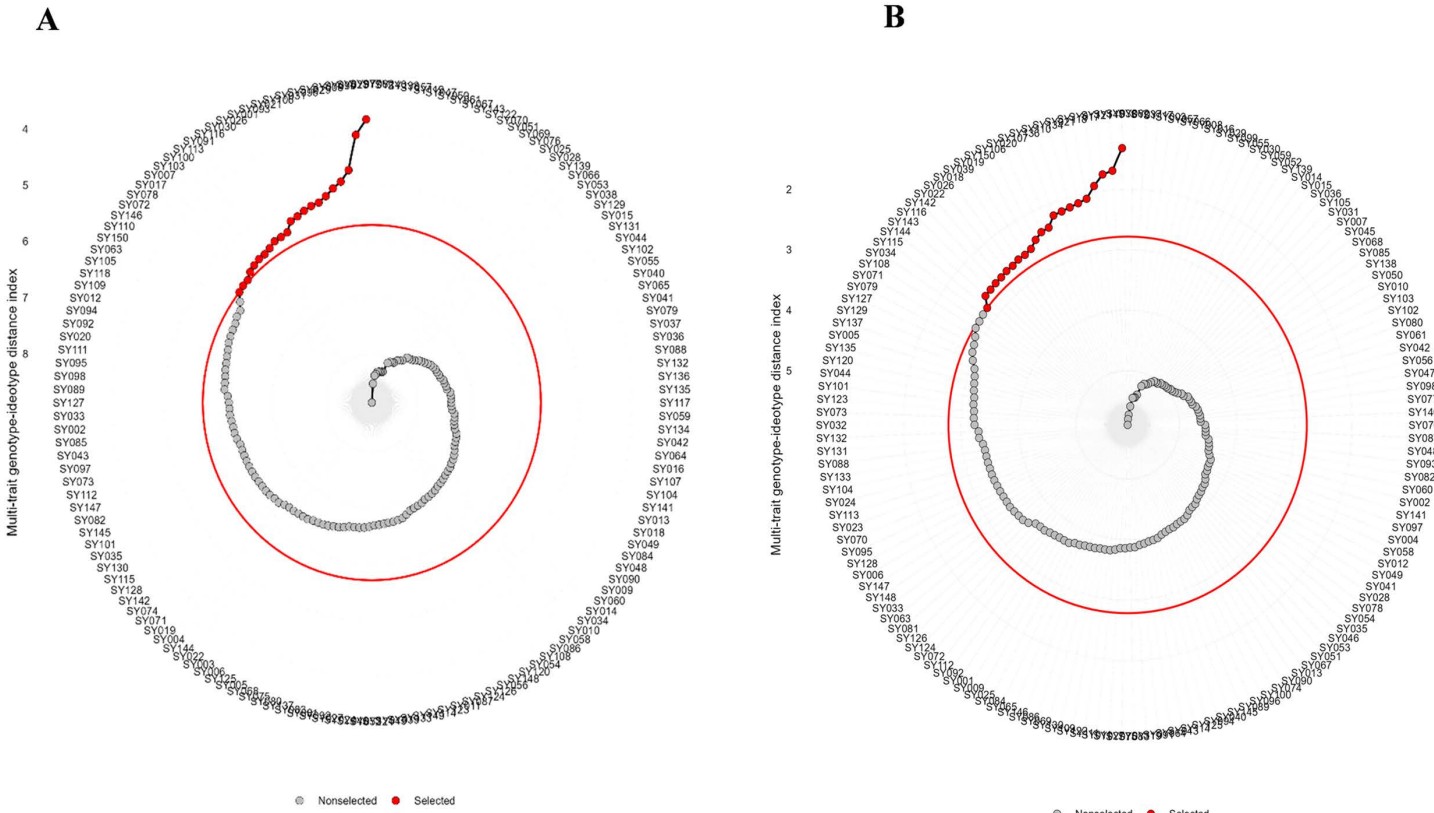

**Fig 3. Views of soybean genotype selection based on the MGIDI index approach.** The selected accessions are depicted as red dots, while the unchosen accessions are shown as black circles. The red circle indicates the cut-off point according to the selection intensity (SI = 20%). (A) Under water-stressed conditions, (B) under well-watered conditions.

**Table 4. Factorial loadings, communalities, uniquenesses and predicted genetic gain based on the MGIDI index of traits measured under water-stressed conditions.**

| Traits | FA1 | FA2 | FA3 | FA4 | Com | Uni | SD (%) | Sense |
|---|---|---|---|---|---|---|---|---|
| D50F | **−0.84** | −0.11 | 0.01 | 0.07 | 0.72 | 0.28 | −6.27 | decrease |
| D95M | **−0.85** | 0.28 | 0.03 | −0.02 | 0.81 | 0.19 | −1.87 | decrease |
| FB | −0.05 | 0.06 | **0.87** | 0.01 | 0.76 | 0.24 | 6.37 | increase |
| PH | −0.06 | **−0.78** | 0.06 | −0.15 | 0.64 | 0.36 | 0.70 | increase |
| NPP | −0.06 | **0.72** | 0.08 | −0.13 | 0.55 | 0.45 | 6.69 | increase |
| NSPP | −0.15 | −0.28 | 0.11 | **−0.70** | 0.60 | 0.40 | 4.62 | increase |
| LS | −0.16 | 0.24 | **−0.73** | −0.02 | 0.61 | 0.39 | −4.68 | decrease |
| HSW | **−0.64** | −0.09 | −0.14 | 0.06 | 0.44 | 0.56 | 3.38 | increase |
| GY | 0.13 | 0.23 | 0.06 | **−0.78** | 0.68 | 0.32 | 46.35 | increase |
| Average | | | | | **0.65** | **0.35** | | |

FA1, FA2, and FA3 = factors 1, 2, 3 and, respectively; Com = communalities; Uni = uniquenesses; SD (%) = predicted selection gains. The bold represents the traits with high contribution to each factor.

**Table 5. Factorial loadings, communalities, uniquenesses and predicted genetic gain based on the MGIDI index of traits measured under well-watered conditions.**

| Traits | FA1 | FA2 | FA3 | Com | Uni | SD (%) | Sense |
|---|---|---|---|---|---|---|---|
| D50F | −0.16 | 0.19 | **−0.73** | 0.59 | 0.41 | −4.60 | decrease |
| D95M | −0.05 | **0.69** | −0.46 | 0.70 | 0.30 | 0.34 | decrease |
| FB | −0.21 | **0.57** | −0.24 | 0.43 | 0.57 | −11.90 | increase |
| PH | **0.82** | 0.03 | 0.04 | 0.68 | 0.32 | 15.75 | increase |
| NPP | −0.23 | −0.03 | **0.74** | 0.60 | 0.40 | −11.21 | increase |
| NSPP | **0.77** | −0.27 | −0.18 | 0.70 | 0.30 | 7.87 | increase |
| LS | **0.26** | −0.07 | 0.24 | 0.13 | 0.87 | −30.10 | decrease |
| HSW | −0.37 | −0.44 | **−0.60** | 0.69 | 0.31 | −1.83 | increase |
| GY | 0.03 | **−0.51** | 0.06 | 0.22 | 0.78 | 0.40 | increase |
| Average | | | | 0.53 | 0.47 | | |

FA1, FA2, FA3, and FA4 = factors 1, 2, 3, and 4, respectively. Com = communality, Uni = uniqueness, SD (%): predicted selection gains. The bold represents the traits that contributed more to each factor.

**A**

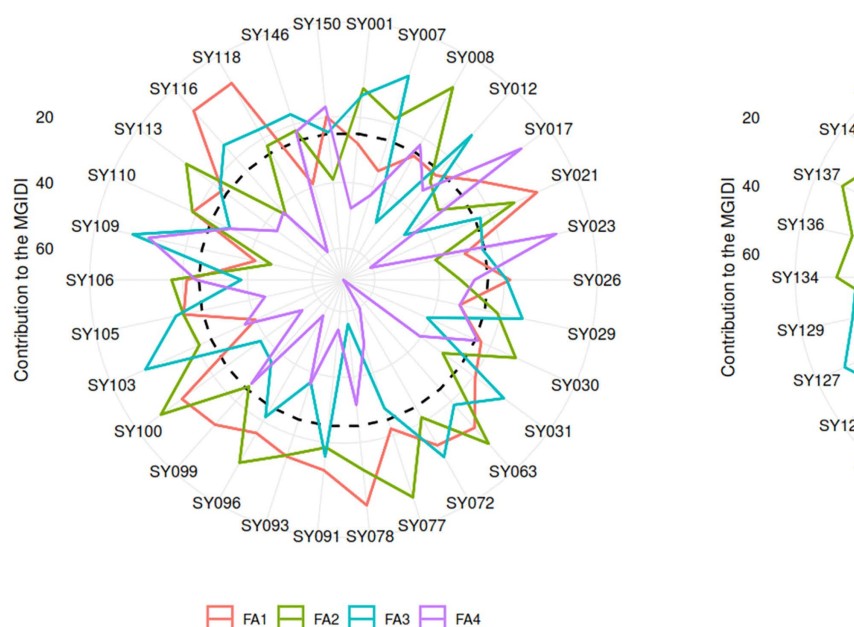

**B**

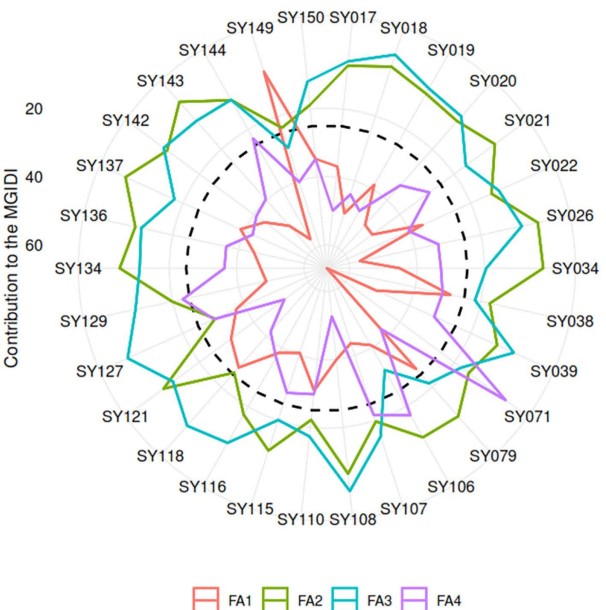

**Fig 4. Radar chart displaying the relative strengths and weaknesses of the selected accessions under water-stressed (A) and well-watered (B) conditions, based on the MGIDI analysis.** The black dashed line at the center represents the theoretical value, assuming equal contributions from all factors.

overlap in selected accessions. Remarkably, genotype SY150 was the only genotype consistently identified by all three methods, suggesting a stable performance across different selection criteria. The analysis further showed that the MGIDI under water-stressed and well-watered conditions shared the highest (eight) number of common accessions. In contrast, the RS ranking method showed greater overlap with MGIDI under water-stressed conditions (five accessions) than with MGIDI under well-watered conditions (three accessions) (Fig 5).

## Discussion

Water stress that occurs during the vegetative stage of the soybean crop can induce minimal changes in plant physiology, morphology, and yield performance once the stress is alleviated [51,52]. However, stress imposed during flowering to early seed setting is critical and tends to have a more substantial negative impact on yield [53,36]. In fact, water stress during flowering is known to cause significant flower abortion, which leads to reduced pod initiation, while stress during pod filling limits nutrient distribution and reduces both seed number and seed weight [40,41]. Therefore, it is crucial to identify, select, or develop soybean cultivars that can tolerate water stress during these sensitive growth stages to maintain productivity and ensure food security [37,38]. The significant differences among the soybean accessions revealed by the combined analysis of variance across each water regime for the studied traits indicate the presence of high genetic variation among these accessions in response to the different water regimes [54]. In a similar study, Mehmood et al. [55] also reported significant differences among the soybean genotypes for various morphological traits, such as days to flowering, days to maturity, plant height, number of grains per pod, hundred-seed weight, and grain yield per plant. The significant mean square effect for the environment main effect for many of the traits indicated that the environment response

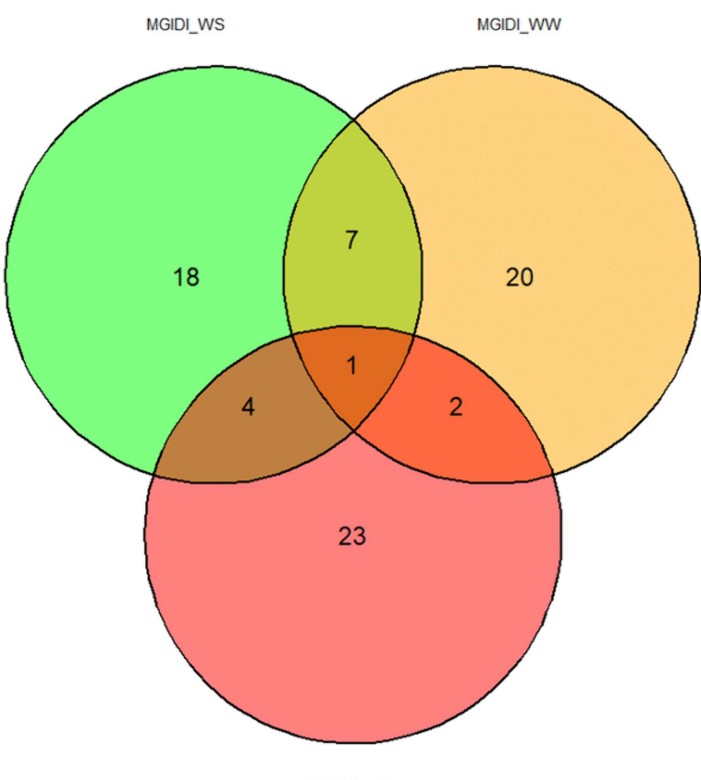

**Fig 5. Venn diagram showing the similarity among the MGIDI methods under both water regimes and the RS selection ranking.**

across the two years of evaluations was not the same. In addition, the significant accessions by environment interaction for grain yield and other key traits indicated the differential response of the soybean accessions to the contrasting water regimes, which also implicates the need for selecting stable accessions that consistently perform well across different water regimes and environmental conditions. Thus, highlighting the importance of selecting stable accessions that consistently perform well across different environments. Environmental factors such as temperature, rainfall, etc., have been reported to play a crucial role in influencing the performances of soybean accessions, suggesting the importance of testing over time and in different environments [56]. These findings emphasize the need for breeding programs to test accessions across multiple environments and watering regimes to ensure yield stability under varying climatic conditions.

The average grain yield under water-stressed conditions was 859 kg/ha, compared to 2324 kg/ha under well-watered conditions. Water stress resulted in an average of 37% reduction in yield. While some soybean accessions were able to minimize yield loss [SY010 (36.8%), SY032 (38.1%), and SY005 (31%)], some showed greater susceptibility potential from their higher yield loss [SY131 (82.5%), SY147 (86.9%)]. Similar yield reductions of 32% and 27.5% among soybean accessions due to water stress were reported by Mathonsi [57] and Zhang et al. [58], respectively. Mathew [59] reported a yield reduction of 40% due to moisture stress in wheat. The average yield under optimum conditions was threefold of the yield under water-stressed conditions, highlighting the potential for identifying high-yield accessions, which could be utilized for improving drought resilience through targeted selection [60].

Broad-sense heritability is a fundamental concept in plant breeding that provides an estimate of the proportion of phenotypic variation in a population that is due to genetic factors. It provides a valuable first look at the genetic control of phenotypic variation, guiding crucial decisions related to selection strategies, population evaluation, breeding method choice, and resource allocation in plant breeding programs. It helps breeders understand the extent of genetic variation and the potential for genetic improvement of soybeans through selection for various desired traits. In drought tolerance studies, breeders use broad-sense heritability to estimate the potential genetic gain that can be achieved through selective breeding for the traits of interest [57]. Our study revealed contrasting patterns of broad-sense heritability ($H^2$) for various agronomic traits in soybean under well-watered and water-stressed conditions. Consistently high $H^2$ estimates were found for days to flowering (D50F), fresh biomass (FB), plant height (PH), number of seeds per plant (NPP) and lodging score (LS) under both water regimes, indicating a strong genetic control over these traits. In contrast, the $H^2$ estimates for days to 95% maturity (D95F), number of seeds per pod (NSPP) and hundred seed weight (HSW) showed a substantial reduction under a water-stressed condition, suggesting a greater influence of environmental factors on the expression of these traits under limited water availability. Given the high heritability estimates observed, effective and reliable selection might be expected for all the studied traits [61]. The higher heritability estimates for traits such as days to 95% maturity (D95F), number of seeds per pod (NSPP) and hundred seed weight (HSW) indicated that these traits can serve as valuable genetic parameters for improving and selecting high-yielding accessions at both environments. These results are consistent with the findings of Talebi & Fayyaz [62], who also reported high heritability for key agronomic traits, such as plant height and number of seeds per spike in wheat, under both water-stressed and well-watered conditions.

Although grain yield (GY) is the primary target in soybean breeding, it consistently showed low heritability under both water regimes in our study. This outcome is expected given the strong sensitivity of yield to environmental variability, particularly under drought, and aligns with earlier findings by Mathonsi [57]. From a genetic gain perspective, low heritability limits the effectiveness of direct selection for yield alone, underscoring the need for alternative strategies. Here, MGIDI provided an advantage by integrating yield with correlated secondary traits, such as days to flowering, number of pods, and seed weight, that display higher heritability. This allows for indirect selection that increases efficiency and reliability in identifying superior genotypes, even when yield heritability is low [63,64].

More broadly, traits with high heritability (>60%) offer greater prospects for consiostent genetic improvement, while low-heritability traits require breeders to account for for strong environmental influence and genotype x environment interactions [46]. For such complex traits, including yield, robust strategies such as multi-environment testing and

marker-assisted or genomic selection are critical to enhance selection accuracy [65–67] Taken together, the results highlight why a multi-trait approach such as MGIDI is particularly valuable cause it levarages the stability of highly heritable secondary traits while still capturing yield potential, thereby addressing the limitations of direct selection under variable environments [68]. [67]. Evaluating soybean accessions under water-stressed and well-watered conditions is crucial for selecting accessions that are tolerant to drought. In this study, drought-tolerant soybean accessions were selected by ranking them based on their adjusted values for each drought tolerance index. Yahoueian et al. [14] have used similar drought tolerance indices to screen 10 soybean genotypes under water stress conditions. Khodarahmpour et al. [69] found that STI and GMP were the more accurate criteria for selecting heat-tolerant and high-yielding maize genotypes. Likewise, Eivazi et al. [70] reported that MP index was the best criterion for selecting barley genotypes with high grain yield under both well-watered and drought-stressed conditions. The polygon views and genotype-by-index interaction analysis revealed strong positive correlations among indices such as STI, GMP, MP, and Ys, confirming their utility for drought tolerance assessment. These findings aligned with Borzoo [71], who also emphasized the effectiveness of MP, GMP, HM and STI for selecting genotypes with superior performance across water regimes. Furthermore, Shojaei et al. [51] and Zendrato et al. [19] also used this approach to identify the most suitable genotypes under stress conditions. This approach ensures the selection of accessions based on each index, producing results comparable to those with the ranking methods. For instance, SY095 and SY032 consistently ranked highly for MP, STI, HM and GMP, while performing less favorably in susceptibility-related indices such as TOL and SSI, highlighting their strong drought tolerance [72].

The significant positive correlation observed between the mean yield under well-watered conditions (Yp) and all drought tolerance indices, except for YSI, is in line with the findings reported by Mathonsi [57]. The positive correlation found between Yp and MP, YI, HM, STI, and GMP indices is in agreement with the findings of Majidi et al. [73], who reported positive and significant correlations between Yp and drought tolerance indices such as TOL, MP, GMP, STI, SSI, and HM. These authors reported that selecting based on indices such as GMP, STI, and HM could enhance yield performance under both stressed and non-stressed conditions. The SSI and TOL indices were found to be significantly and positively correlated, indicating that these indices can possibly predict one another [74,75]. Additionally, Nargeseh et al. [64] and Arif et al. [76] reported a similar positive association between SSI and TOL. The significant positive correlation between grain yield (GY) and each of hundred-seed weight (HSW) and number of seeds per plant (NPP) under water-stressed and well-watered conditions suggests that selecting high-yielding soybean accessions could be achieved through indirect selection for HSW and NPP across both water regimes [77,78].

The multi-trait genotype-ideotype distance index (MGIDI) is an advanced method for genotype screening that incorporates both secondary traits and yield to identify ideal accessions called "ideotypes". This approach helps in selection by computing the distance between the genotype and ideotype as defined by the breeder, which offers a potential solution to challenges such as weighting coefficients for economic traits or addressing multicollinearity issues found in other selection techniques [19,79]. Multi-trait genotype-ideotype distance index (MGIDI) is a valuable tool for identifying accessions with superior mean performance and desired genetic gain while also providing insights on the strengths and weaknesses of selected accessions [27,80]. This method provides a more efficient and precise approach for selecting drought-tolerant soybean accessions by incorporating need-to-have traits rather than relying on single-trait indices [50]. Numerous studies have demonstrated the effectiveness of this method in selecting superior accessions under abiotic stress and across diverse mega-environments [19,24,80,55]. This study used a two-step approach, beginning with multi-trait evaluations under different water regimes, leading to the identification of 30 top-performing accessions per regime. Accessions such as SY017, SY021, SY026, SY106, SY110, SY116, SY118 and SY150 showed consistent excellence across environments, indicating strong potential for breeding. Factor-based grouping emphasized the need to integrate phenotypic, yield-related, and physiological traits in selection. The high predicted selection gain (68.11%) suggests these accessions are promising for drought resilience, particularly those with early flowering and maturity (FA1), strong yield potential (FA4), efficient pod development (FA2), and optimized leaf size for water use efficiency (FA3).

When compared with the drought indices selection method (RS), clear differences emerged. RS primarily emphasized yield under stress conditions, selecting accessions such as SY010, SY032, and SY095. However, many of these accessions lacked consistency across other important traits. In contrast, MGIDI under both water stress (MGIDI-WS) and well-watered conditions (MGIDI-WW) identified accessions such as SY017, SY021, SY026, SY106, SY110, SY116, and SY118, which were not captured by RS despite their desirable balance of early flowering, pod development, and seed size, and yield. These accessions combine both productivity and resilience, ensuring adaptability across contrasting water regimes. Without MGIDI, such multi-trait superior performers would likely have been missed, underscoring how MGIDI complements and strengthens traditional methods by prioritizing stable, ideotype-like accessions [66].

Beyond the immediate results, the findings of this study have broader implications for tropical soybean breeding programs. The MGIDI approach provides an effective strategy to balance yield and resilience in water-limited regions such as sub-Saharan Africa, where irregular rainfall and limited irrigation constrain soybean expansion. This framework reduces dependence on single-trait selection, which often breaks down under variable field conditions.From a practical breeding perspective, this dataset offers two key insights. First, it shows that early-generation selection can be enhanced by integrating secondary traits alongside yield, thereby advancing lines that combine both productivity and resilience. Second, the identification of desired-type accessions using MGIDI provides a foundation for genomic selection, enabling marker effects to be trained on balanced multi-trait datasets rather than yield alone. This strategy is expected to improve prediction accuracy in untested environments and accelerate genetic gain [81].

## Conclusion

This study demonstrated that while traditional drought indices provide useful insights, their reliance on yield alone limits efficiency, given the low heritability and environmental sensitivity of the trait. Based on drought tolerance indices, genotypes SY094, SY010, SY150, SY117, and SY036 were consistently identified as the top drought-tolerant accessions, exhibiting high yield under stress, superior performance across tolerance indices (STI, GMP, HM), and low stress susceptibility. In contrast, genotypes SY070, SY073, SY069, SY129, and SY131 were classified among the most drought-sensitive, characterized by low yield under stress, low tolerance indices, and high susceptibility to drought conditions. These contrasting genotypes represent valuable genetic resources for breeding programs, either as donor parents for drought tolerance or as materials for dissecting the genetic basis of drought response.

By integrating multiple agronomic traits, the MGIDI index proved more robust, identifying eight soybean accessions that combined stable yield with resilience across contrasting water regimes. These findings underscore the value of combining conventional indices with multi-trait approaches to enhance selection accuracy and efficiency. For breeding programs in sub-Saharan Africa and other drought-prone regions, this dual strategy provides a practical pathway to identify high-yielding, resilient candidates for further development. Nonetheless, validation using multi-location and multi-season testing remains essential to confirm the stability and adaptability of the selected accessions. Looking ahead, breeders are encouraged to incorporate MGIDI into early-generation and genomic selection pipelines to accelerate genetic gain under variable environments.

## Supporting information

**S1 File. S1 Table: Soybean accessions used in the study with their respective origins.** S1 Fig. Rainfall pattern under water-stressed and well-watered conditions. S2 Fig. Minimum and maximum temperature pattern under water-stress and well-watered conditions. S2 Table: Description of the traits measured to evaluate the soybean accessions under water stress and well-watered conditions. S3 Table: Grain yield under stress and non-stress conditions and various tolerance indices of the screened soybean accessions. S4 Table: Factor loadings, communalities, uniquenesses and predicted genetic values of the selected accessions under water-stressed conditions based on the multi-trait genotype-ideotype distance index (Bold values represent traits with high contribution to each component). S5 Table: Factor loadings,

communalities, uniquenesses and predicted genetic values of the selected accessions under well-watered conditions based on the multi-trait genotype-ideotype distance index (Bold values represent traits with high contribution to each component).
(ZIP)

## Acknowledgments

A special acknowledgement is due to IITA's Soybean Breeding Units and the Pan African University for the financial and academic support.

## Author contributions

**Conceptualization:** Tenena Silue, Bunmi Olasanmi, Abush Tesfaye Abebe.

**Data curation:** Tenena Silue.

**Formal analysis:** Tenena Silue, Paterne Angelot Agre, Adeyinka Saburi Adewumi, Idris Ishola Adejumobi.

**Funding acquisition:** Abush Tesfaye Abebe.

**Investigation:** Tenena Silue, Paul Olabode Kehinde.

**Methodology:** Tenena Silue, Adeyinka Saburi Adewumi, Idris Ishola Adejumobi, Abush Tesfaye Abebe.

**Project administration:** Abush Tesfaye Abebe.

**Resources:** Abush Tesfaye Abebe.

**Software:** Tenena Silue, Paterne Angelot Agre, Adeyinka Saburi Adewumi, Idris Ishola Adejumobi.

**Supervision:** Bunmi Olasanmi, Paterne Angelot Agre, Abush Tesfaye Abebe.

**Validation:** Tenena Silue, Paterne Angelot Agre, Adeyinka Saburi Adewumi, Idris Ishola Adejumobi, Abush Tesfaye Abebe.

**Visualization:** Tenena Silue, Paterne Angelot Agre, Adeyinka Saburi Adewumi, Idris Ishola Adejumobi, Abush Tesfaye Abebe.

**Writing – original draft:** Tenena Silue.

**Writing – review & editing:** Tenena Silue, Bunmi Olasanmi, Paterne Angelot Agre, Adeyinka Saburi Adewumi, Idris Ishola Adejumobi, Godfree Chigeza, Paul Olabode Kehinde, Abush Tesfaye Abebe.

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
