## [Decision Letter · Decision Letter 0]

1 Sep 2025

Dear Dr. SILUE,

Thank you for submitting your manuscript to PLOS ONE. After careful consideration, we feel that it has merit but does not fully meet PLOS ONE’s publication criteria as it currently stands. Therefore, we invite you to submit a revised version of the manuscript that addresses the points raised during the review process.

We look forward to receiving your revised manuscript.

Kind regards,

Karthikeyan Thiyagarajan, PhD

Academic Editor

PLOS ONE

Journal Requirements:

3. In the online submission form you indicate that your data is not available for proprietary reasons and have provided a contact point for accessing this data. Please note that your current contact point is a co-author on this manuscript. According to our Data Policy, the contact point must not be an author on the manuscript and must be an institutional contact, ideally not an individual. Please revise your data statement to a non-author institutional point of contact, such as a data access or ethics committee, and send this to us via return email. Please also include contact information for the third party organization, and please include the full citation of where the data can be found.

Additional Editor Comments:

Dear Authors,

I appreciate your work concerning the drought tolerant traits evaluation in diverse soybean genotypes under drought stressed and non-stressed conditions. It is better to indicate that whether any genotype or genotypes among the genotypes is a known drought tolerant genotype or cultivar? What is the known drought tolerant soy bean reference cultivar? It would have been better if you could have followed the hill drop method of sowing rather than the drilling method to ensure the fixed distance between the plants. Because soybean is leguminous plant with root nodules in its root, where the rhizobium will fix the atmospheric nitrogen in the form of ammonia. So perhaps the uneven space between the soybean genotypes may experience the interactions from the microbes of rhizosphere as well from its own nodules. So, for your future research, I just suggest to follow a hill dropping kind of sowing method to ensure the regular spacing between the genotypes. Also, there are machineries available to make this process easier.

Some additional suggestions and comments:

Please read this article for the hill drop method of sowing: https://www.scielo.br/j/cr/a/HmWzBpyBdvBsZ4MvcZdxMJj/?lang=en

Is it possible to have a durable abiotic stress resistance?. Because some potential genotypes may not be having a consistency in performance and durability of resistance. Please check this article: https://www.nature.com/articles/s41598-023-28354-0

Have you had a thought of trying a superior genotype over existing varieties in cleistogamous species like Glycine max if you opt through plant breeding and selection?

Dwarf varieties and their genetic, physiological mechanisms and relevant genes, like Rh1, Sd1, etc., were found to have an important role in the green revolution with yield in rice and wheat, for instance. Hence, have you tested any dwarf phenotype varieties? From Table 1, factorial loadings (FA1, FA2, and FA3) with the MGIDI index increase the yield (15.75) with SD: predicted selection gains. How do you infer or compare with reduced plant height if there is a case to study?

According to ANOVA from water stress and well-irrigated conditions, the plant height is not showing significance under the random effect of the environment, thus perhaps due to inherent variations in plant height with specific genotypes. So thus, I suggest selecting specific dwarf and tall genotypes to

test with these water conditions to decipher yield and PH to compare further with your future studies.

You have indicated in the abstract, "Pearson correlation analysis revealed that yields under stress and non-stress conditions showed a moderate positive correlation (r = 0.38), supporting stability across environments." However, it is expected that known drought-tolerant cultivars will have slightly more positive correlations than weak or medium correlations under water stress conditions as well, and it is possible to postulate the hypothesis of having drought-tolerant cultivars with reasonable yields as compared to high-yield lines. How do you infer this?

According to Table 3, there is a huge difference between Yp: yield under well-watered conditions and Ys: yield under water-stressed conditions. But you have mentioned 8 genotypes are consistently performing well in both conditions. What are all those 8 genotypes? Could you please compare them separately with an appropriate table or figure?

Studied genotypes showing the higher broad sense heritability in well-watered conditions over the stress conditions? Do you believe the genotypes would perform better after generations under stress conditions, as H2 is lower in drought conditions? You have also indicated that traits with high heritability (> 60%) indicate strong genetic control and a high possibility of reliable selection from the germplasm of interest.

The correlation figure and other figures have low resolution; please improve them. Please avoid overstating the predicted results, as you have also discussed the weakness of the MGIDI genotype method besides its strength.

Also reviewers provided their comments and suggestions; please address the comments from reviewers and mine and revise the manuscript accordingly.

Reviewers' comments:

Reviewer's Responses to Questions

**Comments to the Author**

1. Is the manuscript technically sound, and do the data support the conclusions?

Reviewer #1: Yes

Reviewer #2: Yes

Reviewer #3: Yes

2. Has the statistical analysis been performed appropriately and rigorously?

Reviewer #1: Yes

Reviewer #2: Yes

Reviewer #3: Yes

3. Have the authors made all data underlying the findings in their manuscript fully available?

Reviewer #1: Yes

Reviewer #2: Yes

Reviewer #3: Yes

4. Is the manuscript presented in an intelligible fashion and written in standard English?

Reviewer #1: Yes

Reviewer #2: Yes

Reviewer #3: Yes

Reviewer #1: Reviewer comments to: PONE-D-25-34303 “Multi-Trait Selection for Drought-Tolerant Soybean Genotypes Under Contrasting Water Regimes” for PLOS ONE

Summary

This paper describes original research in which 150 soybean accessions were subjected to withholding water at two critical points during plant growth, and repeated for two years. Performance of these accessions was also measured during well-watered conditions. The goal was to identify germplasm tolerant to water deficit stress. This paper validates multi-trait selection (MGIDI) which is not a new idea, but might be unique for soybeans in Nigeria, subjected to water deficit. The authors might add several sentences to clarify the scientific novelty of this study.

In general the experiment was conducted with appropriate methods, and statistical procedures. Sufficient experimental material was used. In general, the study was described in sufficient detail, although the soil used should be described (see note 4). The study identified 30 accessions with better tolerance of water deficit, and 8 accessions that were better in a range of conditions. No single trait best described plant water stress tolerance, and required multiple traits to identify suitable accessions.

Revision of the manuscript is suggested.. with attention to correct use of ‘drought’ (see note 1), and ‘genotype’ (see note 2). Other points are noted below.

Page numbers refer to the number listed for the manuscript body (rather than the collated PDF)

Main points

1) P1 L2 (also P5 L102-104, and throughout the manuscript) Avoid using water stress and drought stress interchangeably. Drought is variation from climatic normality. Since your water deficit is applied artificially, it is better to use the term ‘water stress’, or ‘water deficit stress’ (to distinguish from flooding stress). Drought, from below average rainfall, is likely to have seasonal duration, and might vary from the short-term water deficits used in this study. Drought is also typically associated with other climatic factors such as higher temperature and/or lower humidity, as well as water deficit. In this study, short-term water deficit is a tool for determination of accession performance, and likely is related to drought tolerance, however, that was not measured in this study.

2) P1 L3 title (also P1 L16, P1 L21, P3 L70, and throughout the document). Ensure that genotype is only used to describe a single genetic identity. Avoid the colloquial use of ‘genotype’ to describe a ‘line’, or ‘accession’. In most instance, ‘genotype’ can be replaced with ‘germplasm’. Table S1 in Supplementary materials describes the germplasm being used as ‘accessions’. Its not clear in this paper if the accessions were genotypes (i.e. inbred or clonal germplasm), but for soybeans, its likely these were selections from outcrossed germplasm, comprising many genotypes. Although there are precedents for colloquial use pf ‘genotype’ in the literature, “accessions” should not be referred to as ‘genotypes’, unless these are inbred lines comprising a single genotype.

3) P3 L48-49. Revise this sentence; as overly simplistic. Drought is infinitely variable, so giving a single yield loss is misleading. Drought can reduce soybean yield by 100%!!

4) P4 L89. Describe the soil used for the study.

5) P5 L100. Presumably, the replications were whole blocks?

6) P5 L111. Clarify ‘Nodumax’. Was this a commercial product. Presumably is rhizobia?

Specific points

7) P1 L18 (also P5 L105). ‘Drought’ is a noun, and avoid colloquial use as an adjective. Suggest to replace ‘drought stress’ with ‘water stress’

8) P1 L78. Use past tense ‘aimed’. L79. Replace ‘are’, with ‘were’. L86. Replace ‘will proved’, with ‘provided’

9) P3 L56. Add ‘growth’, i.e. ‘… during critical growth stages…’

10) P4 L86. Use singular ‘gain’ (how many ‘genetic gains’ are you describing here?)

11) P4 L96. Avoid the possessive term. Suggest “The climate of Ibadan is …”

12) P5 L107. Avoid the possessive term. Suggest “plant”

13) P5 L113. Replace “and that gives a total of”, with “equivalent to”

14) P5 L114-116. Use lower case ‘metribuzin, and ‘urea’. Check journal style, do these chemicals need their full chemical name at first mention?

15) P17 L313. Use singular ‘Multi-trait’

16) P17 L316 (also L411). Suggest to replace ‘Under’, with “During”

17) P17 L322. Use past tense ‘was’

18) P20 L389. Changes in what?

19) P20 L391. Delete ‘economic’. Drought will simply reduce yield.. not need to include the term ‘economic’

20) P20 L397. Replace ‘like’, with ‘such as’ (L438-439 is correct)

21) P23 L451. Use past tense ‘indicated’

Reviewer #2: The drought tolerance of soybean genotypes or cultivars varies during the reproductive stage. Assessing the differences among varieties, as well as understanding the relationship between their phenotypic and agronomic traits and drought tolerance, and their contribution to yield, is of practical significance. This study comparatively analyzed the drought tolerance of 150 soybean genotypes and screened for drought-tolerant genotypes. However, there are three issues: (1) When different genotypes are planted in various regions with differing environments and soils, they may exhibit distinct characteristics. Therefore, field trials typically employ multi-point evaluations; however, it remains to be seen whether the drought-tolerant genotypes identified in one region can be applicable in other agro-ecological zones. (2) While several methods were used to compare drought tolerance among genotypes, the study does not specify which method is more representative for assessing drought tolerance in soybeans. (3) The manuscript is well-written and clear, but it could be further refined.

Reviewer #3: General Comments:

The authors used multi-trait selection indices, particularly MGIDI, in combination with traditional drought tolerance indices, across two contrasting water regimes. This strengthen the screening and selection efficiency. The manuscript is generally well-structured, with a clear rationale and thoughtful statistical methods.

However, some methodological clarifications, results interpretation improvements, and stronger framing of novelty are needed before publication.

• Several grammatical errors and non-scientific phrases are present throughout.

• Intensive English language revision is required.

• Use consistent verb tense—prefer past tense for methods and results.

Title & Abstract

• The abstract is long and verbose. Consider tightening the language by focusing more on outcomes than method repetition.

• Mention how this research advances the field by demonstrating MGIDI's use for drought selection in tropical soybean germplasm.

Introduction

• A clear articulation of the knowledge gap is lacking. What specifically is not known or has not been done in similar germplasm or environments?

• Please include information that clearly stating:

o Why previous selection strategies were insufficient.

o What novelty MGIDI + traditional indices bring in combination.

o Why the IITA germplasm is uniquely suitable for this kind of selection.

• A clear hypothesis is missing.

Materials and Methods

• There is no control/check varieties. Without benchmark genotypes, it's difficult to interpret relative performance or derive conclusions about breeding progress.

• Are the two-stage drought simulation realistic stress?

• Temperature and rainfall variations across the years/seasons are critical in drought studies.

• Add a climatic data figure comparing environmental conditions in WW and WS seasons.

• The field management description lacks clarity on soil properties, nutrient status, and irrigation uniformity. These are critical for attributing differences in genotypic response to water regime rather than confounding effects.

• Two row per plot is not experimentally viable.

• What was the kind of sprinkler irrigation system? Model? Etc.

• It’s not clearly stated whether the experimental unit was plot-based or individual plant-based for statistical analysis.

• Clarify unit of replication and randomization protocol used within blocks. Was spatial variation corrected?

• The rationale for timing and duration needs justification using prior physiological data on soybean water stress sensitivity. How did you choose these two time points?

Statistical Analysis

• While explained mathematically, lacks the MGIDI implementation explanation for breeders.

• Include a statement on whether model assumptions were checked and met.

Results & Discussion

• Grain yield (GY) had low H² under both conditions; this warrants deeper discussion, especially since it's the main selection target. Is MGIDI offset low GY heritability? This contradicts genetic gain theory unless indirect selection traits are used.

• Ensure each figure clearly labels axes, index names, and genotype IDs.

• Add a paragraph in discussion, comparing genotypes selected by MGIDI vs. traditional indices (beyond SY150). What kind of genotypes would have been missed without MGIDI?

• The discussion focuses heavily on confirming existing findings (e.g., GY drops with drought), but misses a chance to generalize implications for other tropical soybean breeding programs.

• What can breeders take away from this dataset? How could this inform early-stage breeding or genomic selection?

Conclusion

• Slightly descriptive.

• Add a concise final paragraph with a call to action.

• No validation is provided for selected genotypes. Without validation, conclusions on stability remain speculative.

.

Reviewer #1: No

Reviewer #2: **Yes:**Hou LongyuHou LongyuHou LongyuHou Longyu

Reviewer #3: **Yes:**Arshad JalalArshad JalalArshad JalalArshad Jalal

---

## [Decision Letter · Decision Letter 1]

11 Jan 2026

Dear Dr. SILUE,

Thank you for submitting your manuscript to PLOS ONE. After careful consideration, we feel that it has merit but does not fully meet PLOS ONE’s publication criteria as it currently stands. Therefore, we invite you to submit a revised version of the manuscript that addresses the points raised during the review process.

We look forward to receiving your revised manuscript.

Kind regards,

Diaa Abd El-Moneim

Academic Editor

PLOS One

Journal Requirements:

Reviewers' comments:

Reviewer's Responses to Questions

**Comments to the Author**

Reviewer #2: All comments have been addressed

Reviewer #4: (No Response)

2. Is the manuscript technically sound, and do the data support the conclusions?

Reviewer #2: Yes

Reviewer #4: Partly

3. Has the statistical analysis been performed appropriately and rigorously?

Reviewer #2: Yes

Reviewer #4: Yes

4. Have the authors made all data underlying the findings in their manuscript fully available?

Reviewer #2: Yes

Reviewer #4: Yes

5. Is the manuscript presented in an intelligible fashion and written in standard English?

Reviewer #2: Yes

Reviewer #4: Yes

Reviewer #2: The revised manuscript has addressed my questions , and I have no further questions.

Reviewer #4: Reviewer Comments to Author:

Ms. Ref. No.: PONE-D-25-34303R1

Ms. Full Title: Multi-Trait Selection for Drought-Tolerant Soybean Accessions Under Contrasting Water Regimes

The authors conducted a study looking at the performance and tolerance level of 150 soybean genotypes from IITA collection when exposed to drought. The study carried out across two growing seasons use multi-traits selection approach under both well-watered and water-stressed condition. This is an important study. However, clarifications and amendments are to be made before the paper to be considered for publication in this valued journal

Methods:

[1]. Provide the characteristics of the soil used for testing the tolerance. What is the field capacity (FC) of that soil? What is the Permanent wilting Point (PWP)?

[2]. You have to explain each formula and the meaning of each index. Does the increase of each of these indices is leading to the plant tolerance?

[3]. You stressed the plant twice: At 35 DAS for 14 days, corresponding to the flowering stage and at 63 DAS for 7 days, corresponding to the pod-filling stage. Before combining both stresses, using each stress independently would have allow the comparison the stress induced during the flowering stage and the one made during the pod filling stage and coming out with the most sensitive stage between the flowering and the pod filling phases. What did you not consider independent stress?

Results:

[1]. Are the results presented in Table 3 are for the 30 best performing genotypes under stress? It is the results for which year? Are they for the combined two years? as the experiment was carried out during two consecutive years

[2]. Results should also present the least performing genotypes under stress, the most sensitive accessions which also of significant importance for a breeder, who can use them for example in crossing when looking the type of gene action for the tolerance to drought

Conclusion

[1]. In the conclusion, It would be interesting to name the top (Example Top 5) genotypes tolerant to drought and also the top genotypes sensitive to drought

.

Reviewer #2: No

Reviewer #4: **Yes:**Eric Bertrand KouamEric Bertrand KouamEric Bertrand KouamEric Bertrand Kouam

---

## [Author Response · Author response to Decision Letter 2]

27 Jan 2026

Dear Reviewer

We sincerely thank the Editor and reviewers for their careful evaluation and insightful comments. All suggestions have been thoroughly considered and addressed, and the manuscript has been revised to enhance its clarity, scientific rigor, and overall quality. We believe that the revised manuscript adequately responds to all concerns raised and is substantially improved as a result.

---

## [Decision Letter · Decision Letter 2]

23 Feb 2026

Multi-Trait Selection for Drought-Tolerant Soybean Accessions Under Contrasting Water Regimes

PONE-D-25-34303R2

Dear Dr. SILUE,

We’re pleased to inform you that your manuscript has been judged scientifically suitable for publication and will be formally accepted for publication once it meets all outstanding technical requirements.

Kind regards,

Diaa Abd El-Moneim

Academic Editor

PLOS One

Additional Editor Comments (optional):

Reviewers' comments:

Reviewer's Responses to Questions

**Comments to the Author**

Reviewer #4: All comments have been addressed

2. Is the manuscript technically sound, and do the data support the conclusions?

Reviewer #4: Yes

3. Has the statistical analysis been performed appropriately and rigorously?

Reviewer #4: I Don't Know

4. Have the authors made all data underlying the findings in their manuscript fully available?

Reviewer #4: Yes

5. Is the manuscript presented in an intelligible fashion and written in standard English?

Reviewer #4: Yes

Reviewer #4: Substancial improvements have been carried out, allowing the manuscrpt suitable for publication in this valued journal

.

Reviewer #4: **Yes:**Eric Bertrand KouamEric Bertrand KouamEric Bertrand KouamEric Bertrand Kouam

---

## [Editor Report · Acceptance letter]

PONE-D-25-34303R2

PLOS One

Dear Dr. SILUE,

I'm pleased to inform you that your manuscript has been deemed suitable for publication in PLOS One. Congratulations! Your manuscript is now being handed over to our production team.

Kind regards,

on behalf of

Dr. Diaa Abd El-Moneim

Academic Editor

PLOS One